# DMA: Enhancing Retrieval-Augmented Generation with Adaptive Human Feedback

## Abstract

Retrieval-augmented generation (RAG) systems typically rely on static retrieval methods, limiting their adaptability to dynamic environments. In this paper, we propose a novel online learning framework called Dynamic Memory Alignment (DMA), designed specifically to enhance retrieval performance and content generation in RAG through adaptive incorporation of multi-level human feedback. DMA systematically integrates real-time feedback signals at document, list, and response levels, effectively adjusting memory management strategies to optimize relevance and adaptability in online interactive environments. Extensive evaluations demonstrate DMA's competitive foundational retrieval performance across multiple standard knowledge-intensive benchmarks. DMA achieves significant improvements on datasets reflecting natural conversational interactions (TriviaQA, HotpotQA), confirming its suitability for online GenAI dialogue applications. Moreover, a multi-month industrial deployment demonstrates that DMA substantially improves user engagement in real-world applications. These results underscore DMA's ability to maintain robust foundational retrieval capabilities while excelling at dynamic, real-time adaptation in interactive online environments.

## 1 Introduction

Retrieval-augmented generation (RAG) has become a core paradigm for enhancing the factuality and adaptability of LLMs in knowledge-intensive tasks (Lewis et al., 2020; Borgeaud et al., 2022). By decoupling parametric memory from non-parametric retrieval, RAG enables models to access external information dynamically, grounding responses on up-to-date and domain-specific knowledge without modifying internal parameters. This separation has powered recent advances across open-domain QA (Izacard & Grave, 2021), multi-hop reasoning (Yang et al., 2018), and instruction-based augmentation (Lin et al., 2024; Gao et al., 2023).

Despite these advances, conventional RAG pipelines exhibit critical limitations in dynamic online settings: (i) Static retrieval strategies cannot adapt to evolving user intent or content drift. Most dense retrievers are trained offline and remain fixed at deployment time, failing to reflect live interaction signals (Lin et al., 2023; Jiang et al., 2024). (ii) Given the limited context length of mainstream LLMs (Liu et al., 2023), retrieval must prioritize highly relevant information. Sole reliance on top-$k$ dense similarity often results in suboptimal recall and necessitates robust re-ranking strategies (Nogueira et al., 2020; Glass et al., 2022b; Qin et al., 2024). (iii) While dedicated rankers and hybrid retrievers can improve retrieval precision (Ma et al., 2023; Izacard et al., 2022), they often lack the flexibility and generalization needed for personalized, real-time adaptation (Zhang et al., 2024a). These issues collectively suggest that current RAG systems require an adaptive interface between user feedback and memory control.

Motivated by these challenges, our goal is to build an adaptive online learning framework for RAG systems that effectively integrates and utilizes dynamic human feedback, enabling continuous real-time refinement of memory and retrieval decisions. Recent studies demonstrate that instruction-tuned LLMs can effectively align responses with user intent through task-specific fine-tuning (Liu et al., 2024b; Lin et al., 2024). Real-time human feedback across document-, list-, and response-level granularity can serve as actionable supervision signals for adaptive retrieval. DMA incorporates these signals through continuous feedback-driven memory alignment.

To this end, we propose Dynamic Memory Alignment (DMA), an innovative online learning framework designed to systematically organize, interpret, and incorporate adaptive human feedback signals, dynamically optimizing retrieval strategies and memory prioritization within RAG workflows.

Specifically, DMA addresses the core challenge of online adaptability through three key components: (1) a multi-granularity feedback taxonomy tailored for conversational GenAI scenarios; (2) a suite of reward modeling techniques that interpret heterogeneous user signals into structured supervision; (3) online knowledge fusion mechanisms that prioritize high-value memory traces and modulate retrieval policy accordingly.

As a result, the DMA framework is particularly suited to real-time, user-facing applications such as chat assistants and enterprise QA bots, where system adaptability is key to sustained performance (Asai et al., 2024b; Jeong et al., 2024).

Our contributions can be summarized as follows:

- We propose DMA, a novel online learning framework enabling RAG systems to continuously refine adaptive retrieval based on multi-level user feedback. DMA systematically captures sparse yet valuable user signals to dynamically enhance system responsiveness in dynamic online settings.

- Through extensive evaluations on widely-used knowledge-intensive benchmarks, DMA achieves strong results on conversational datasets such as TriviaQA and HotpotQA, showing state-of-art performance than prior leading methods.

- Most critically, DMA demonstrates notable real-world applicability, as evidenced by a 24.57% improvement in positive user feedback during a multi-month randomized controlled industrial trial, validating its effectiveness and adaptability in practical deployment.

The remainder of this paper is structured as follows: § 2 surveys related work. § 3 formalizes the RAG problem and highlights key limitations of static pipelines. § 4 presents the proposed DMA framework. Experimental setup and results are detailed in § 5, while remaining challenges are discussed in § 6. We conclude in § 7.

## 2 Related Work

RAG has emerged as a core solution for knowledge-intensive NLP tasks (Lewis et al., 2020; Borgeaud et al., 2022). In standard RAG pipelines, a dense retriever (e.g., (Karpukhin et al., 2020)) encodes queries and documents into a shared embedding space, retrieving top-$k$ relevant contexts from an external corpus. These retrieved contexts are then fused with the input query and processed by an LLM to generate grounded responses (Izacard & Grave, 2021; Izacard et al., 2023).

Recent research has focused on enhancing this pipeline along several directions. One thread optimizes retrieval to better align with the downstream generation needs of LLMs (Shi et al., 2024; Lin et al., 2024; Ye et al., 2023). Another line introduces multi-step and interleaved retrieval-generation mechanisms to capture complex reasoning chains (Trivedi et al., 2023; Shao et al., 2023; Jeong et al., 2024). Meanwhile, context filtering and selection strategies have been developed to remove noisy evidence before generation (Wang et al., 2023; Xu et al., 2024; Yoran et al., 2024), improving both factuality and efficiency.

In parallel, instruction tuning has become a critical enabler for aligning LLMs with retrieval-enhanced tasks. From supervised instruction collections like FLAN and Self-Instruct (Wei et al., 2022; Wang et al., 2022) to open-source alignment efforts such as ChatGPT and Claude (OpenAI, 2023; Anthropic, 2023), LLMs increasingly learn to operate over retrieved evidence. Recent studies demonstrate that retrieval-augmented instruction tuning significantly boosts performance across QA and reasoning tasks (Liu et al., 2024b; Asai et al., 2024b; Lin et al., 2024; Luo et al., 2023; Wang et al., 2024).

Nevertheless, integrating retrieval into LLM training remains challenging due to the need for surrogate losses and continuous re-indexing (Guu et al., 2020; Shi et al., 2024; Sachan et al., 2021; Izacard et al., 2023; Dong et al., 2024).

Ranking-based enhancements have been extensively used to improve retrieved context quality before generation. Early neural ranking models (Mitra et al., 2018; Chen et al., 2020) were later extended to dual-stage architectures such as Re2G (Glass et al., 2022b), PARADE (Drozdov et al., 2023), and RA-DIT (Lin et al., 2024), enabling more flexible reordering. However, these rankers often rely on moderate-sized encoder models (e.g., BERT or T5), which struggle with complex semantics and generalization (Ram et al., 2023). Recent evidence suggests that full-scale LLMs can act as powerful rankers with minimal prompting (Qin et al., 2024; Sun et al., 2023; Khalifa et al., 2023), yet leveraging this capacity in online RAG systems remains under-explored.

Crucially, most prior work optimizes retrieval and re-ranking on static datasets, assuming fixed user intent and corpus distribution. This paradigm fails to accommodate the non-stationary dynamics in real-world online systems, where user behavior, topic drift, and feedback evolve continuously. To bridge this gap, emerging approaches such as Self-RAG (Asai et al., 2024a), ReFeed (Yu et al., 2024), and Pistis-RAG (Bai et al., 2024) propose adaptive mechanisms incorporating implicit or explicit feedback. These methods are typically confined to limited settings and do not offer general-purpose integration into end-to-end retrieval and memory control.

In contrast, the DMA framework introduces a unified online learning architecture that encodes multi-level user feedback at document-, list-, and response-level granularity into dynamic retrieval optimization. Our approach maintains continuous feedback loops to enable retrieval and generation components to co-adapt during deployment, which supports sustained performance in open-ended, user-facing GenAI systems.

# 3 Preliminaries

This section formalizes the RAG pipeline that serves as the foundation for our work. We then identify key limitations of existing RAG approaches in dynamic online settings, which motivate the design of our proposed DMA framework.

## 3.1 Problem Setup

Let $\mathcal{C} = \{d_1, d_2, \ldots, d_N\}$ denote a corpus of external knowledge documents. Given a user query $q \in \mathbb{Q}$, a retriever $R$ computes similarity scores using dense embeddings, typically in a dual-encoder setting (Karpukhin et al., 2020), where $\text{Relevance}(q, d_i) = \langle E_q(q), E_d(d_i) \rangle$ and $E_q, E_d$ are the query and document encoders. The top-$k$ documents are selected as $D_{\text{retrieve}} = \text{Top}_k\{\text{Relevance}(q, d_i) \mid d_i \in \mathcal{C}\}$. A reranker $\text{Rerank}_m$ may be applied to reorder and truncate this list to the top-$m$ items, yielding $D = \text{Rerank}_m(q, D_{\text{retrieve}})$, where $m \leq k$ (Cao et al., 2007; Glass et al., 2022a). The final context set $D = \{d^{(1)}, \ldots, d^{(m)}\}$ is concatenated with the query and fed into a language model $G$ to generate a grounded response $a = G(q, D)$. While the retriever $R$ and generator $G$ may be trained separately or jointly (Sachan et al., 2021; Izacard et al., 2023), most real-world systems adopt modular training due to scalability and deployment constraints.

## 3.2 Limitations of Current Approaches

Despite their success in open-domain question answering and related tasks (Lewis et al., 2020; Borgeaud et al., 2022; Guu et al., 2020), current RAG systems exhibit structural limitations when deployed in dynamic, user-facing environments.

First, conventional RAG methods rely on static retrievers trained offline over frozen corpora, using task-specific training signals (e.g., NQ, TriviaQA) that do not generalize well to continuously evolving user needs (Lewis et al., 2020; Guu et al., 2020; Izacard et al., 2023). This fixed retrieval logic fails to accommodate domain drift, long-term user preferences, or topic shifts typical of online applications.

Second, although mechanisms such as reranking or filtering (Chen et al., 2020; Wang et al., 2023; Xu et al., 2024) can improve precision, they are typically rule-based or learned from fixed supervised data. These components rarely leverage live user feedback signals, and even when available, such signals are often aggregated in limited forms (e.g., binary preference) or only utilized post hoc.

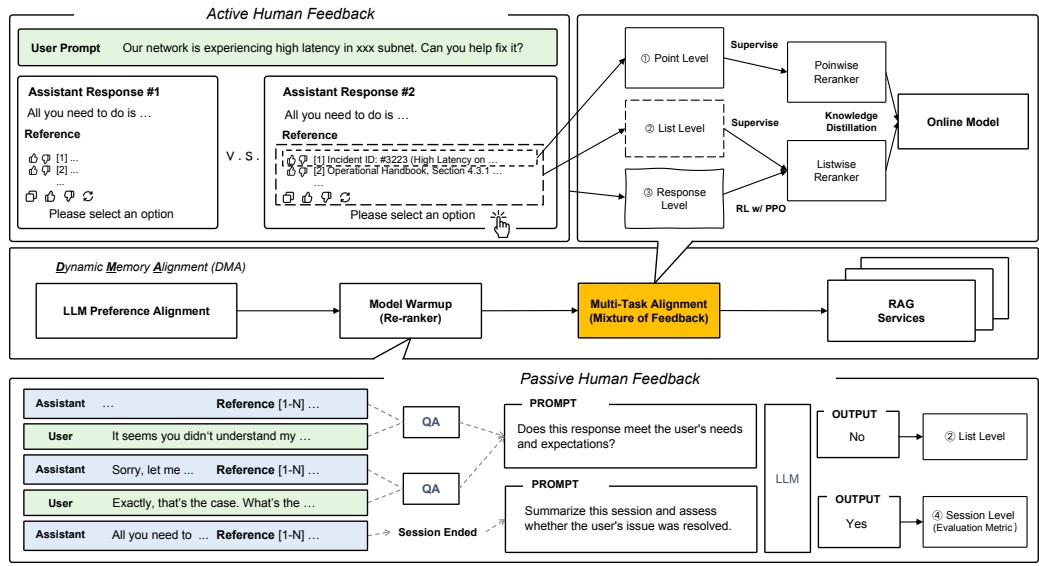

Figure 1: Overview of the DMA feedback loop. Multi-level human feedback is organized, modeled, and fused to guide online retrieval strategies. Reranker training and distillation are detailed in Figure 2.

Third, most RAG systems lack a principled framework to incorporate multi-granular feedback—such as document-level usefulness, list-level coverage, or response-level satisfaction—into real-time retrieval decisions. While reinforcement learning from human feedback (RLHF) (Ouyang et al., 2022) and browser-based systems such as WebGPT (Nakano et al., 2021) have demonstrated the potential of fine-grained supervision, these approaches remain decoupled from the retrieval components and are difficult to generalize to streaming environments.

As a result, retrieval behavior remains largely fixed during deployment, limiting the system's ability to improve with usage, personalize to users, or adapt to shifts in content distribution. These limitations call for an online learning mechanism capable of dynamically integrating human feedback into memory and retrieval policies—precisely the gap that our proposed DMA framework aims to address.

## 4 Dynamic Memory Alignment

To address the limitations mentioned in the previous section, we introduce **DMA**, an online learning framework designed to continuously refine retrieval strategies in RAG systems by leveraging real-time user feedback. Unlike conventional static pipelines, DMA forms a closed-loop system that adaptively aligns memory and retrieval decisions with evolving user preferences.

### 4.1 Framework Overview

As illustrated in Figure 1, DMA comprises three core components: (1) **Feedback Taxonomy**, which structures heterogeneous user signals into well-defined levels; (2) **Reward Modeling**, which transforms these signals into trainable supervision; and (3) **Online Adaptation**, which updates retrieval strategies based on real-time feedback. Together, these modules form a dynamic feedback loop, enabling memory alignment in continually evolving GenAI interactions.

### 4.2 Human Feedback Taxonomy

Effective capture and utilization of user feedback in industrial settings require systematic organization. Addressing the challenge of sparse and heterogeneous feedback across user contexts, we investigate prominent LLMs, including ChatGPT (Achiam et al., 2023), Gemini (Team et al., 2023), QWen (Bai

162 et al., 2023), DeepSeek (Liu et al., 2024a), ChatGML (GLM et al., 2024), and Kimi (Team et al.,
163 2025), to identify and categorize various forms of human feedback signals.

164 This taxonomy provides a systematic structure to interpret diverse feedback signals and optimize
165 system behavior. As illustrated in Figure 1, feedback signals are categorized into four levels of
166 granularity:

167 **1) Document-level feedback** reflects user evaluations of individual retrieved snippets, typically
168 through direct actions such as upvoting or downvoting. This feedback is formalized into a preference
169 dataset $\mathcal{D}_{\text{pref,doc}} = \{(q_i, d_i, y_{q_i,d_i})\}_{i=1}^{N}$, enabling optimization of document-level relevance.

170 **2) List-level Feedback** captures user preferences over a set of retrieved documents, evaluating the
171 overall quality of system outputs for a query $q_i$ based on a list $D_{q_i}$ and system response $y_{q_i}$. This
172 includes both explicit (e.g., copy, regenerate) and implicit feedback. It is formalized into a preference
173 dataset $\mathcal{D}_{\text{pref,list}}$, providing insights into document relevance and ranking consistency for a list subset
174 $D_{\text{sub},q_i}$.

175 **3) Response-level feedback** refers to user preference between two (or more) response options
176 generated from distinct document sets $D_1$ and $D_2$. The feedback signal $y$ indicates the pre-
177 ferred response, implying a preference between the document sets. This is formalized as $\mathcal{D}_{\text{resp}} =$
178 $\{(q_i, r_{1,i}, r_{2,i}, D_{1,i}, D_{2,i}, y_i)\}_{i=1}^{N_{\text{resp}}}$. This data is valuable for alignment methods and can be scaled.

179 Although the feedback is collected at the response level, each response is generated based on a
180 specific document list. As such, user preference over responses implicitly reflects preference over the
181 underlying document sets, which we leverage to supervise document-level reranking.

182 **4) Session-level feedback** aggregates user evaluations across an entire interaction session $s_i$, capturing
183 overall user perceptions such as task satisfaction $f_{s_i}$. While this high-level signal is not used directly
184 to train granular reward models, it is employed in two key roles: (i) as an external metric for evaluating
185 DMA variants (§5.1); and (ii) as a dynamic weight signal to adjust fusion importance across feedback
186 types during GBDT distillation (see §4.3) (Friedman, 2001).

187 By structuring feedback into these levels and formalizing the associated datasets, our taxonomy offers
188 a robust framework for systematically interpreting user inputs and optimizing GenAI systems.

## 4.3 Reward Construction and Memory Alignment

190 To leverage the multi-granular feedback captured by the taxonomy for optimal DMA performance,
191 we design specific modeling methods for each granularity level and develop strategies to combine
192 their outputs to influence memory alignment. A multi-task modeling approach integrates diverse
193 feedback signals to construct reward signals suitable for training memory alignment components.

194 **Document-Level Modeling.** A model is trained using explicit document-level feedback $\mathcal{D}_{\text{pref,doc}}$
195 with Binary Cross-Entropy (BCE) loss $\mathcal{L}_{\text{BCE}} = -y \log \sigma(s) - (1 - y) \log(1 - \sigma(s))$, where $s$ is
196 the predicted score and $y \in \{0, 1\}$ is the label. This produces a pointwise reranker focused on
197 fine-grained precision.

198 **List-Level Modeling.** To capture relative importance among retrieved results, listwise rerankers are
199 trained using user feedback. The ListNet loss $\mathcal{L}_{\text{ListNet}} = -\sum_i P_{\text{true}}(i) \log P_{\text{pred}}(i)$, where $P(i) =$
200 $\exp(s_i)/\sum_j \exp(s_j)$, ensures alignment between predicted and target ranking distributions.

201 **Response-Level Modeling.** We collect pairwise user feedback comparing responses $(r_1, r_2)$ gen-
202 erated from different document lists $D_1, D_2$, forming preference data $\mathcal{D}_{\text{resp}} = \{(q, D_1, D_2, y)\}$
203 with binary preference label $y$. A reward model $R(D)$ is trained via pairwise loss $\mathcal{L}_{\text{pairwise}} =$
204 $-y \log \sigma(R(D_1) - R(D_2)) - (1 - y) \log \sigma(R(D_2) - R(D_1))$, where $\sigma(\cdot)$ is the sigmoid func-
205 tion. To inject response-level preferences into the reranker, we apply Proximal Policy Optimization
206 (PPO) (Schulman et al., 2017), optimizing a listwise policy using the clipped surrogate objective
207 $\mathcal{L}_{\text{PPO}} = -\mathbb{E}_t[\min(r_t \widehat{A}_t, \text{clip}(r_t, 1 - \epsilon, 1 + \epsilon)\widehat{A}_t)]$, where $r_t = \pi_\theta(a_t|s_t)/\pi_{\theta_{\text{old}}}(a_t|s_t)$ and $\widehat{A}_t$ is the
208 advantage estimated from the reward model. This approach effectively aligns a listwise reranker
209 using response-level feedback, producing the **PPO-aligned listwise reranker**, which captures global
210 user satisfaction signals beyond document or list-level heuristics.

**Fusion and Distillation for Online Serving** To meet the latency requirements of online serving, we distill the outputs of upstream feedback-supervised rerankers into a lightweight ensemble model. Specifically, we adopt a Gradient Boosting Decision Tree (GBDT) as the final online scoring module.

This GBDT model is trained using soft labels derived from upstream reranking components and provides efficient inference without sacrificing alignment quality. In production deployment, it enables real-time document ranking with sub-10ms latency while preserving the benefits of multi-granular feedback supervision.

Figure 2 illustrates the high-level training pipeline. Additional modeling and supervision details are omitted for brevity and deployment sensitivity.

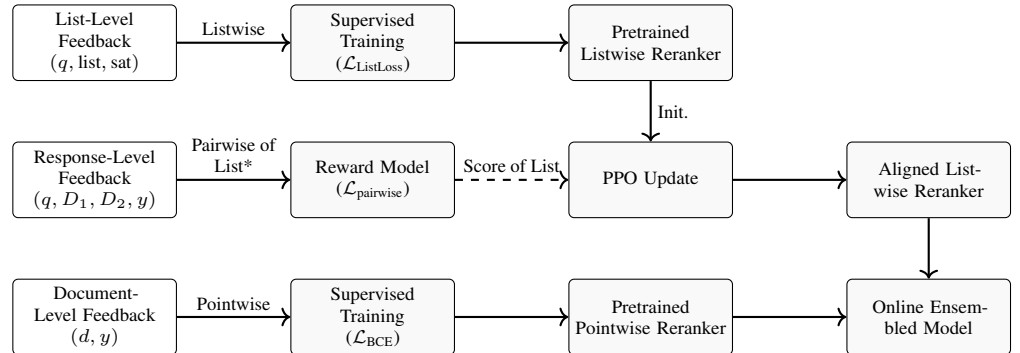

Figure 2: Training and distillation architecture in DMA. Three types of feedback supervise specialized ranking models, which are fused via PPO and distilled into a lightweight online reranker. Full pipeline details in § 4.3

## 5 Experiment

We evaluate DMA across two settings: (1) real-world online interactions with online users, and (2) public open-domain QA benchmarks. The former validates DMA's online learning ability in production; the latter assesses generalization under static evaluation protocols.

### 5.1 Experiment Setup

**Evaluations on API Distribution.** We conduct a multi-month randomized controlled trial (RCT) on a Chinese-language GenAI system operated by a major telecommunications and cloud provider. To support multilingual retrieval, DMA uses BGE-m3 as the retriever backbone and an instruction-tuned decoder for response generation.

To characterize domain diversity, session queries are categorized into seven application areas: Technical support (37%), Performance and monitoring (21%), API and developer support (16%), Security and compliance (10%), Service and resource management (9%), Migration and deployment (4%), and Product features and updates (3%). The query distribution reflects the industrial and technically specialized nature of the evaluation environment.

For measurement, we define session-level satisfaction as $S(s_i) = \frac{1}{n_i} \sum_{j=1}^{n_i} \mathbb{I}(\text{LLM}(q_{i,j}, y_{i,j}) \neq$ dissatisfied$)$, where each session $s_i$ contains $n_i$ user turns. Because explicit user ratings are sparse, we employ QWen2-72B (Yang et al., 2024) as an automated annotator to infer satisfaction labels, calibrated via in-context few-shot learning using high-quality examples from online human feedback.

To ensure alignment with human judgment, these labeled sessions are treated as ground-truth supervision during prompt construction. The prompt template (Table 1) specifies the annotator's role, input-output format, and exemplar completions. The structured outputs include a categorical label (satisfied, neutral, or dissatisfied), a confidence score in $[0, 1]$, and a short list of improvement suggestions.

This automated feedback serves as the primary evaluation signal for DMA under real-world usage. Inter-annotator agreement analysis confirms high label reliability, with a Cohen's Kappa of 0.962 between model predictions and human annotations.

Table 1: Session-level User Satisfaction Evaluation Prompt Design

| Intent | Prompt |
|---|---|
| **Role** | You are an AI assistant responsible for evaluating user satisfaction at the session level. |
| **Task** | Assess overall user satisfaction based on the entire conversation history, including user queries and system responses. |
| **Input** | A session $s_i$ consisting of $n_i$ turns: $\{(q_{i,j}, y_{i,j})\}_{j=1}^{n_i}$, where $q_{i,j}$ is the user query and $y_{i,j}$ is the system-generated response. |
| **Few-shot Examples** | $\{$`few_shot_examples`$\}$ illustrating different types of session outcomes. |
| **Output Format** | **User Satisfaction**: `satisfied` / `neutral` / `dissatisfied` 
 **Confidence**: A numerical value in $[0, 1]$ representing model confidence. 
 **Improvements**: A short list of suggestions to improve the user experience. |

**Evaluations on public static benchmarks.** To evaluate generalization in static settings, we test DMA on four standard open-domain QA datasets: Natural Questions (NQ: 79.2k train / 8.7k dev / 3.6k test) (Kwiatkowski et al., 2019), TriviaQA (78.8k / 8.8k / 11.3k) (Joshi et al., 2017), HotpotQA (88.9k / 5.6k / 5.6k) (Yang et al., 2018), and WebQSP (2.8k / 250 / 1.6k) (Berant et al., 2013). These benchmarks span a range of query types, from open-ended to structured factoid-style tasks. We report Hit@1 and F1 following prior work. To ensure comparability with existing methods. All generations were performed using a unified LLaMA2-7B decoder (Touvron et al., 2023), controlling for decoding variability and isolating retrieval alignment effects.

**Implementation Details.** DMA's online update pipeline is triggered after accumulating 500 new feedback samples using Flink-based monitoring. This threshold was empirically selected to balance the need for timely adaptation against the computational cost of frequent retraining. It ensures that model updates are based on sufficient feedback to generate stable gradient signals, while also preventing excessive latency in high-throughput environments. In practice, this results in update intervals ranging from several minutes to an hour, depending on traffic volume.

To accommodate variable traffic conditions, the feedback monitoring system automatically defers updates if insufficient feedback is collected, avoiding retraining on sparse or noisy signals. This adaptive scheduling ensures robustness across deployment scales, from high-traffic production environments to slower-feedback applications.

The full DMA update cycle includes: (1) training pointwise and listwise teacher models, (2) generating soft distillation targets, and (3) training a 10K-tree GBDT model. Over 90% of the latency is spent on teacher model training ($\approx$ 6 minutes) and distillation ($\approx$ 3 minutes), with model checkpoint updates taking less than 1 minute. The system runs on 8 A800 GPUs per training job, yielding an average end-to-end update latency of 10 minutes (range: 6–15 minutes). To maintain sub-15-minute updates as feedback volume grows, GPU capacity is scaled proportionally. Online response generator QWen2-72B (Yang et al., 2024) is served via vLLM (Kwon et al., 2023) to support high-throughput inference. Feedback events are streamed through Apache Flink pipelines.

## 5.2 Main Results

We evaluate DMA in two complementary settings: a multi-month industrial deployment to assess its real-world effectiveness in large-scale online environments, and four public QA benchmarks to verify its retrieval and generation performance under standard static protocols.

**Results on Real-World Online Evaluation.** As shown in Table 2, Full DMA yields a 24.57% increase in session-level user satisfaction over an online BGE-based reranker baseline. This improvement is statistically significant ($p < 0.001$, two-tailed z-test), based on 100,000 user sessions collected via a randomized controlled trial. For detailed results on the impact of different feedback signals and alignment strategies, see § 5.3.

Table 2: User satisfaction across four evaluation settings. (A) compares DMA against a static baseline (BGE-Reranker). (B) reports the effect of removing individual feedback signals from DMA. (C) analyzes fusion strategies. (D) compares online learning to weekly batch updates.

| Configuration | User Satisfaction (%) | Relative Change (%) |
|---|---|---|
| **(A) Overall Performance** | | |
| Zero-Aligned reranker (baseline) | 62.11 | Reference |
| Full DMA (ours) | **77.37** | +24.57 |
| **(B) Feedback Ablation** | | |
| Full DMA (baseline) | 77.37 | Reference |
| w/o List-Level Feedback | 65.32 | −15.57 |
| w/o Response-Level Feedback | 68.70 | −11.21 |
| w/o Document-Level Feedback | 73.29 | −5.27 |
| **(C) Fusion Strategy** | | |
| Cascading Fusion (baseline) | 72.79 | Reference |
| Distillation (Full DMA) | **77.34** | +6.25 |
| **(D) Online Learning** | | |
| Weekly Batch Learning (baseline) | 76.21 | Reference |
| Online Learning (Full DMA) | **77.54** | +1.75 |

*Impact of Fusion Strategy.* To evaluate the performance of our model fusion strategies at scale, we compare distillation against a cascading approach using the online RCT setup. Table 2 shows that distillation outperforms cascading by +6.25% under similar latency constraints.

*Impact of Online Learning.* User preferences evolve over time, necessitating continuous model updates. We evaluate the impact of DMA's online learning mechanism, which performs incremental daily retraining and real-time feedback adaptation, compared to a baseline of weekly batch updates. As shown in Table 2, online learning improves session-level satisfaction by +1.75% compared to batch learning, providing qualitative evidence for the value of continuous adaptation.

Table 3: **Results on Public QA Benchmarks Grouped by Task Type.** Left: Conversational QA datasets (open-ended user queries). Right: Structured QA datasets (schema-grounded queries). All methods are evaluated using LLaMA2-7B as the reader model, which serves as the largest publicly available common denominator across prior work to ensure fair and standardized comparison.

| Method | Conversational QA Tasks | | | | Structured QA Tasks | | | |
|---|---|---|---|---|---|---|---|---|
| | TriviaQA | | HotpotQA | | NQ | | WebQSP | |
| | Hit@1 | F1 | Hit@1 | F1 | Hit@1 | F1 | Hit@1 | F1 |
| KnowPAT (Zhang et al., 2023) | 63.20 | 65.20 | 29.00 | 37.40 | 51.42 | 54.82 | 68.73 | 65.31 |
| RRHF (Yuan et al., 2023) | 62.50 | 60.20 | 28.16 | 35.40 | 50.11 | 52.01 | 66.90 | 63.10 |
| RAFT (Zhang et al., 2024b) | 60.10 | 57.40 | 30.20 | 35.80 | 50.24 | 53.86 | – | – |
| FILCO (Wang et al., 2023) | 67.30 (2) | 67.80 (2) | 32.70 (2) | 40.80 (2) | **52.71** (1) | **55.32** (1) | **69.96** (1) | **68.34** (1) |
| DMA (Ours) | **68.81** (1) | **68.90** (1) | **33.92** (1) | **41.88** (1) | 51.11 (3) | 54.92 (2) | 67.26 (3) | 65.03 (3) |

**Results on Public QA Benchmarks.** To evaluate DMA under standardized retrieval conditions, we assess its performance on four widely used public datasets: TriviaQA (Joshi et al., 2017), HotpotQA (Yang et al., 2018), NQ (Kwiatkowski et al., 2019), and WebQSP (Berant et al., 2013). These span open-ended (TriviaQA, HotpotQA) and schema-grounded (NQ, WebQSP) query types, supporting analysis of generalization across formats. We compare against several alignment-optimized RAG baselines, including KnowPAT (Zhang et al., 2023), RRHF (Yuan et al., 2023), RAFT (Zhang et al., 2024b), and FILCO (Wang et al., 2023). This selection balances method comparability (all adopt alignment-based RAG optimization), result availability (publicly reported scores), and experimental fairness (standardized decoding with LLaMA2-7B (Touvron et al., 2023)). As shown in Table 3, DMA achieves the highest Hit@1 and F1 scores on conversational datasets, and remains competitive on structured tasks. These results underscore DMA's advantage in open-ended, user-facing QA scenarios.

## 5.3 Ablation Studies

We conduct ablation studies to assess the contribution of each feedback granularity in DMA. Table 2 shows that removing *list-level feedback* results in the largest performance drop (–15.57%), followed by *response-level* (–11.21%) and *document-level feedback* (–5.27%). This validates our design choice to integrate multi-granular feedback.

**Hierarchical impact of feedback types.** These results reveal a natural hierarchy in feedback utility: list-level signals provide coarse but globally informative supervision for document ranking; response-level feedback reflects downstream user preferences across document sets; and document-level labels offer fine-grained, local guidance. Their removal leads to progressively degraded satisfaction, confirming their complementary roles.

**Complementarity and alignment.** Pointwise (document) signals alone are insufficient for ranking complex lists, while listwise and response-level supervision offer stronger alignment with holistic user intent. This stack of feedback levels enables DMA to optimize both local document quality and global retrieval behavior, especially in dynamic online environments.

**Takeaway.** Among all components, listwise feedback plays the most critical role in guiding DMA toward globally aligned memory selection. Our multi-granularity design not only enhances overall quality but also ensures adaptability to diverse user preferences in real-world deployments.

## 6    Limitations

While DMA demonstrates robust performance across both public datasets and industrial deployments, two practical limitations remain when applying the framework to broader scenarios:

**Scalability in low-resource or interface-constrained environments.** DMA is designed for large-scale, high-throughput production systems where continuous user feedback is available for online adaptation. In low-traffic or offline settings, feedback signals may be too sparse to support timely model updates. DMA relies on multi-level behavioral signals such as document-, list-, and response-level feedback, which are primarily available in interactive dialogue systems. In structured API-style tasks or static document editing scenarios, such fine-grained feedback is either unavailable or hard to instrument, limiting DMA's adaptability. To mitigate this, DMA includes an adaptive retraining scheduler that defers updates under low-feedback conditions, and future work may explore synthetic or proxy signals to fill these gaps.

**Generalization to schema-bound QA benchmarks.** Although DMA achieves strong results on open-ended, user-facing datasets (e.g., TriviaQA, HotpotQA), its gains are less pronounced on schema-constrained tasks such as NQ and WebQSP. These datasets often feature fixed entity-relation structures or short factual queries that benefit less from multi-granular reranking or feedback-driven adaptation. In such settings, static retrievers and minimal re-ranking may already suffice. This suggests that DMA's dynamic memory alignment is most beneficial in open-ended or conversational environments, and additional strategies—such as symbolic augmentation or knowledge graph integration—may be required to improve performance on interface-like or structured retrieval tasks.

## 7    Conclusion

We present DMA, an online learning framework that systematically incorporates multi-level human feedback (document, list, and response) to enable real-time retrieval alignment in RAG systems. DMA enables adaptive memory selection guided by user preferences, addressing the rigidity of static retrieval pipelines.

DMA achieves state-of-the-art performance on QA tasks and demonstrates significant gains in a large-scale industrial RCT. Its adaptive scheduling and fusion strategies ensure robustness and efficiency, while ablation studies highlight the importance of feedback granularity in performance gains.

Future work will explore extensions to low-resource domains, alternative feedback modalities, and real-time interpretability for memory selection. Overall, DMA demonstrates that structured user interaction signals can powerfully guide online retrieval learning in deployed GenAI systems.

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
