# OpenReview forum: "DMA: Enhancing Retrieval-Augmented Generation with Adaptive Human Feedback"
_NeurIPS.cc/2025/Conference — Submitted to NeurIPS 2025_

### Official Review · Reviewer_NAmN · 2025-06-25

**Clarity:** 1
**Significance:** 2
**Originality:** 3
**Rating:** 3
**Confidence:** 4

**Summary:**

This paper presents Dynamic Memory Alignment (DMA), a novel online learning framework for enhancing Retrieval-Augmented Generation (RAG) systems. The key innovation lies in integrating multi-level human feedback—spanning document, list, and response levels—into the continuous adaptation of both retrieval and reranking modules. By systematically capturing and modeling real-time user preferences, DMA aims to address the inherent rigidity of static retrieval pipelines. The authors demonstrate the framework’s efficacy through extensive experiments, including a large-scale industrial randomized controlled trial (RCT) and evaluations on multiple public QA benchmarks. Results show significant gains in user satisfaction and competitive performance relative to state-of-the-art baselines.

**Questions:**

1. Could the authors provide concrete examples of what constitutes "implicit feedback" at the list level? Additionally, what heuristics or modeling strategies are used to transform these signals into actionable training labels?
2. How exactly is the ListNet loss (Line 199) computed? Specifically, how are the ground-truth relevance scores for the target distribution $P_{\text{true}}$ derived from user feedback? Are they heuristic, probabilistic, or model-predicted?
3. The paper states that BGE-m3 is used as the retriever backbone (Line 227), but what architectures are used for the pointwise and listwise rerankers? Are they also BGE-based, or do they use different encoder models?
4. Could the authors formally define the state space, action space, and the reward function used in the PPO-based listwise reranker training (Lines 205–210)? This is critical for understanding the RL framing and reproducibility.

**Ethical Concerns:**

["NO or VERY MINOR ethics concerns only"]

**Final Justification:**

I acknowledge that the paper proposes a potentially valuable approach to addressing limitations in existing RAG systems. However, the original manuscript lacked critical content and suffered from unclear descriptions, which significantly detracted from its overall quality. While the rebuttal outlines concrete steps to resolve these issues, the scope of the necessary revisions remains substantial. At this stage, it is difficult to definitively evaluate the quality of the revised manuscript. In light of these considerations, I have adjusted my score from 2 to 3.

**Limitations:**

yes

**Paper Formatting Concerns:**

No concern

**Quality:**

2

**Strengths And Weaknesses:**

### Strengths
1. The paper tackles a critical limitation in existing RAG systems: their inability to adapt to evolving user intent and content drift due to static retrieval mechanisms. DMA's design for real-time, feedback-driven adaptation offers a meaningful advancement.
2. The paper introduces a well-structured taxonomy for modeling feedback at three levels—document, list, and response—ensuring holistic alignment between retrieval decisions and user intent.
3. The authors provide rigorous empirical evidence, including industrial-scale deployment results and strong performance on public QA benchmarks, demonstrating both practical viability and generalizability of the proposed method.

### Weaknesses
1. Several critical components of the DMA framework lack adequate description:
  - The paper briefly mentions that list-level feedback incorporates both explicit and implicit signals (Line 172) but fails to define what constitutes implicit feedback or how it is transformed into training-ready preference labels.
  - The knowledge distillation process from upstream rerankers to the final GBDT model (Lines 211-213) remains under-explained. Key questions about soft label generation and GBDT training procedures (loss function, input features, etc.) remain unanswered.
  - Important implementation details are missing, including hyperparameters for the rerankers and PPO optimization, the exact number of retrieved documents (k), and architectural specifics of the reranking models.
2. The paper suffers from several notational ambiguities that hinder readability:
  - The symbol R is overloaded, used for both the retriever (Line 116) and the reward model (Line 203).
  - The symbol y refers inconsistently to a binary preference label (Line 169), a generated system response (Line 171), and a preference over responses (Line 176). Similarly, symbols a (Line 122) and r (Line 178) both denote responses in different contexts.
  - Figure 2 introduces terms like “sat” and (*) without clear explanation.
  - The first feedback level is called “Point Level” in Figure 1 but “Document-level” in the main text (Line 167), causing confusion.
  - Additionally, all formulas are presented inline, making the presentation of key objectives like the PPO loss difficult to follow. Block-level math formatting would substantially improve clarity.
3. Some conclusions in the paper would benefit from stronger quantitative backing:
  - The “Takeaway” section in the ablation study (Lines 316–318) provides only qualitative observations about the relative importance of feedback levels. This discussion would be more convincing if supported by quantitative metrics or a more detailed breakdown of ablation results.
  - The paper would also benefit from more concrete case studies or qualitative examples showing how DMA adapts to user feedback in specific, realistic scenarios.

---

> ### Author Rebuttal · Authors · 2025-07-28
>
> We thank Reviewer NAmN for one of the most thoughtful and constructive reviews we received. We appreciate your recognition of our core contributions—including the novelty of the problem, the structured feedback taxonomy, and the rigor of our empirical evaluation.
>
>
> Summary of Contributions
>
> To briefly reframe:
>
> 1. We introduce a multi-level feedback taxonomy (document, list, response) to capture fine-grained user signals.
> 2. We propose a two-stage online adaptation framework, combining listwise learning (ListNet) and PPO fine-tuning with a distillation layer.
> 3. We validate our system via a large-scale industrial RCT, offering strong real-world evidence and a deployable blueprint for adaptive RAG.
>
>
> Group 1: Methodological Clarifications
>
> Two-Stage Listwise Training (ListNet + PPO)
>
> * Stage 1: Train with ListNet using global feedback (e.g., thumbs-up/down on final response), treating the associated document list as a ranked unit.
> * Stage 2: Fine-tune using PPO with rewards from a pairwise-trained reward model (RM). Preferences between response pairs are projected back to document lists; PPO adjusts the reranker accordingly. The full pipeline is shown in Figure 2.
>
> Q1. Implicit Feedback
>
> Implicit feedback refers to user sentiment inferred from replies. We classify it using a prompted LLM with Cohen’s Kappa = 0.962 (vs. human annotations), applying a confidence threshold of 0.8.
>
> Action: We will add a dedicated subsection describing this pipeline.
>
> Q2. ListNet Loss and Ground Truth
>
> Ground-truth scores are heuristically derived from final response-level ratings (e.g., thumbs-up). We apply ListNet loss over the associated document list.
>
> Action: We will formalize this derivation and present the loss as block-level math.
>
> Q3. Reranker Architecture
>
> * Retriever: BGE-m3.
> * Baselines: Online baseline is static BGE-reranker; public dataset comparisons use known rerankers.
>
> Action: We will clarify this in *Implementation Details*.
>
> Q4. PPO MDP Definition
>
> * State: User query + retrieved document list
> * Action: Reranked document list (permutation)
> * Reward: RM score for the reranked list
>
> Action: We will add this MDP formulation to the methodology.
>
> GBDT Distillation + Hyperparameters
>
> Action: We will include a new appendix with full details on the distillation pipeline (Flink-based), feature set, and all key hyperparameters (e.g., k, PPO settings).
>
>
> Group 2: Notation and Presentation
>
> We agree with the issues raised and will make the following corrections:
>
> * Symbol Disambiguation: Clarify overloaded symbols (`R`, `y`, etc.).
> * Terminology Fixes: Replace “Point Level” with “Document-level” throughout for consistency.
> * Equation Formatting: Convert all major equations to block-level math for clarity.
> * Figures: Revise all terminology and add definitions to figure captions.
>
>
> Group 3: Strengthening Analysis
>
> Ablation and Case Studies
>
> While the original manuscript included ablations, the presentation lacked quantitative clarity.
>
> Action: We will expand the analysis to include numerical metrics showing the impact of each feedback level. We will also add more concrete case studies (before/after examples) in the appendix to illustrate DMA's adaptive behavior.
>
>
> We believe the technical foundation of our work is strong, and your comments have provided a clear roadmap to improve clarity and completeness. We are committed to incorporating all revisions and are confident they will result in a significantly improved manuscript.

---

> ### Comment · Reviewer_NAmN · 2025-08-03
>
> Thank you for your detailed response addressing my concerns. I acknowledge that the paper proposes a novel and potentially valuable approach to addressing limitations in existing RAG systems. However, the original manuscript lacked critical content and suffered from unclear descriptions, which significantly detracted from its overall quality. While your rebuttal outlines concrete steps to resolve these issues, the scope of the necessary revisions remains substantial. At this stage, it is difficult to definitively evaluate the quality of the revised manuscript. In light of these considerations, I have adjusted my score.

---

> > ### Author Response · Authors · 2025-08-04
> >
> > Dear Reviewer NAmN,
> >
> > Thank you for your re-evaluation and your detailed final comments. We acknowledge your valid point that the manuscript in its current form lacks sufficient detail on several critical components.
> >
> > As we briefly mentioned in the submission, we intended to provide these comprehensive details in an appendix and the expanded camera-ready version, given the strict page limits of the initial submission. Your feedback has provided us with a clear and valuable roadmap on which aspects require the most thorough elaboration.
> >
> > We wish to reaffirm our commitment to implementing all revisions outlined in our rebuttal. The final manuscript will be substantially expanded to include:
> >
> > A formal MDP definition for the PPO framework.
> >
> > A detailed walkthrough of the GBDT knowledge distillation process.
> >
> > A complete list of hyperparameters and architectural specifics for reproducibility.
> >
> > We are confident that the revised, complete manuscript will fully address your concerns regarding clarity and reproducibility. Thank you once again for your constructive guidance, which will significantly improve the quality of our paper.
> >
> > Sincerely,
> >
> > The Authors of Submission 8481

---

### Official Review · Reviewer_DVFp · 2025-06-30

**Clarity:** 2
**Significance:** 3
**Originality:** 3
**Rating:** 3
**Confidence:** 3

**Summary:**

The paper addresses a critical limitation of conventional Retrieval-Augmented Generation (RAG) systems: their reliance on static retrieval strategies, which hinders adaptability to dynamic user intents or evolving content in real-time applications (e.g., chatbots). To solve this, the authors propose Dynamic Memory Alignment (DMA), a novel online learning framework that continuously optimizes RAG retrieval through multi-level human feedback. DMA bridges the gap between static RAG pipelines and dynamic user environments by establishing a closed-loop system where retrieval strategies evolve via real-time feedback, enabling sustained performance in open-ended, user-facing GenAI applications.

**Questions:**

1. Why does DMA underperform on schema-bound QA (e.g., NQ, WebQSP in Table 3) despite excelling in conversational tasks? Is this due to feedback bias, architecture limitations, or data distribution?

2. What exact input features (e.g., embeddings, ranking scores) and fusion methods (e.g., weighted logits, multi-loss) are used for GBDT distillation?

3. Can core innovations (e.g., feedback scheduling, distillation) be validated without proprietary infrastructure/data?

4. How do you address feedback-induced biases (e.g., popularity bias → filter bubbles) beyond §6’s theoretical discussion?

5. DMA’s performance tied to massive private LLMs (QWen-72B)? What are the tradeoffs with smaller/OSS models (e.g., Llama-3-70B)?

**Ethical Concerns:**

["NO or VERY MINOR ethics concerns only"]

**Final Justification:**

The author's rebuttal addresses my concerns to some extent; however, without more experimental results, I cannot improve my score.

**Limitations:**

yes

**Quality:**

2

**Strengths And Weaknesses:**

Strengths:
1. Novel hierarchical modeling (document/list/response/session) with specialized objectives: BCE (document), ListNet (list), PPO+reward modeling (response), and session-guided fusion.

2. Industrial-Scale Validation: 24.57% ↑ user satisfaction in 100k-session RCT and SOTA on TriviaQA/HotpotQA provide strong empirical backing.

3. Propose a novel framework to jointly optimize document-, list-, and response-level signals via PPO+distillation.

Weaknesses:
1. About the limited generalization analysis:
- Schema-Bound QA Underperformance: Competitive but not SOTA on NQ/WebQSP (Table 3). Why does DMA lag in entity-centric tasks? Insufficient discussion.
- Feedback Sparsity Mitigation: "Adaptive scheduling" (§5.1) defers updates but lacks low-resource solutions (e.g., synthetic feedback).

2. About deployment Ethics and over-personalization risks:
- Bias Amplification: Human feedback may reinforce retrieval biases (e.g., prioritizing popular documents). §6 mentions "echo chambers" but offers no guardrails.
- Privacy Tradeoffs: Anonymization protects users but obscures demographic fairness analysis (e.g., satisfaction gaps across user groups)

3. Deployment on proprietary infrastructure (telecom provider) and unreleased code/data limit independent validation.

---

> ### Author Rebuttal · Authors · 2025-07-28
>
> Response to Reviewer DVFp
>
> We thank Reviewer DVFp for the thoughtful and insightful review. We greatly appreciate your accurate understanding of the DMA framework, as well as your recognition of its key contributions—including hierarchical feedback modeling, PPO-based alignment, GBDT distillation, and large-scale deployment. Your questions address central aspects of our system, and we are pleased to clarify them below.
>
>
> Summary of Contributions (for clarity)
>
> To briefly reframe, our work proposes a deployable framework for real-world retrieval-augmented generation (RAG) via:
>
> 1. Multi-level Human Feedback Taxonomy
>    A unified schema for organizing document-, list-, and response-level feedback, enabling fine-grained supervision and preference modeling in dynamic online settings.
>
> 2. Online Adaptation Pipeline
>    A scalable learning loop combining reward modeling, listwise PPO alignment, and GBDT distillation, triggered continuously by live user signals.
>
> 3. Industrial-Scale Validation
>    A months-long randomized controlled trial in a high-traffic GenAI system, offering rare real-world evidence of DMA’s impact under production constraints.
>
>
> Q1. Why does DMA underperform on schema-bound QA tasks?
>
> Clarification: This is an expected outcome based on task characteristics. DMA is designed for open-ended, user-facing QA tasks—such as TriviaQA and HotpotQA—where user preference signals (e.g., document utility, list quality, response satisfaction) are rich and nuanced.
>
> In contrast, schema-bound QA tasks (e.g., NQ, WebQSP) have singular factual answers and little tolerance for subjective variation. As such, the multi-granular preference modeling central to DMA contributes less value in these settings.
>
> Action: We will explicitly discuss this design trade-off and task alignment rationale in the *Analysis* (§5.3) and *Limitations* (§6) sections.
>
>
> Q2. What are the GBDT input features and fusion methods?
>
>
> Clarification: DMA employs a two-stage hybrid retraining strategy designed to balance long-term stability with short-term adaptability:
>
> * (1) Weekly Full Retraining: A high-confidence corpus of implicit feedback (satisfaction score ≥ 0.8, derived via Qwen-72B with few-shot calibration) is accumulated over time. Once per week, a fresh base model is trained on this full corpus to absorb slow distributional drift and correct long-term biases.
>
> * (2) Nearline Incremental Updates: On top of the weekly base, the system continuously accumulates \~500 new feedback instances via a Flink-based streaming pipeline. Upon reaching the threshold, the following update cycle is triggered:
>
>   * Train teacher models using document-, list-, and response-level feedback.
>   * Fuse supervision via PPO into an aligned reranker.
>   * Distill into a 10K-tree GBDT for low-latency deployment (<10 ms).
>   * Deploy within \~10 minutes (on 8×A800 GPUs), ensuring rapid adaptation.
>
> Action: We will add a dedicated subsection in §5.1 (*Experimental Setup*) to describe this hybrid retraining architecture, including batch triggers, feedback routing, latency breakdown, and deployment intervals.
>
> Q3. Can DMA be validated without proprietary infrastructure?
>
> Yes. To ensure full reproducibility and support further research, we will release:
>
> 1. Full Codebase: Including reward models, PPO alignment, GBDT distillation, and pipeline orchestration.
> 2. Pretrained Models & Scripts: For TriviaQA, HotpotQA, and other public benchmarks.
> 3. Synthetic Feedback Simulator: A tool for generating document-, list-, and response-level feedback to support offline experimentation with DMA.
>
> These assets allow researchers to study DMA’s behavior and capabilities independently of proprietary data or infrastructure.
>
>
> Q4. How does DMA mitigate feedback-induced bias?
>
> DMA incorporates several safeguards to control both feedback noise and systematic bias:
>
> 1. Confidence Filtering: Implicit signals are retained only if confidence ≥ 0.8, as estimated by the calibrator model (Qwen2-72B).
> 2. Controlled Training Role: High-confidence implicit feedback is used exclusively in weekly full retraining, not in nearline adaptation, preventing overreaction to transient signals.
> 3. Global Distribution Normalization: During training, we apply resampling and normalization strategies (e.g., batch norm, loss weighting) to balance overrepresented patterns and reduce reward hacking risk.
>
> Action: We will expand §6 to discuss these mechanisms and add an ablation analysis in the Appendix quantifying the impact of confidence thresholding on performance stability.
>
>
> Q5. Is DMA tied to proprietary models like Qwen-72B?
>
> Clarification: No. While we use Qwen-72B in production due to its multilingual performance, DMA is fully model-agnostic. The architecture is modular and can be applied to any decoder (e.g., LLaMA, ChatGLM) and calibrator.
>
> In fact, the central innovation of DMA lies in its feedback processing and adaptation pipeline, not in the choice of base LLM. The framework supports plug-and-play integration with arbitrary generation models.
>
> Action: We will clarify this generality in the *Discussion* section, along with practical guidance for adapting DMA to alternative LLMs.
>
>
> We sincerely thank you again for your thoughtful feedback and constructive suggestions. Your comments will directly improve the clarity, reproducibility, and positioning of our final manuscript.

---

> > ### Comment · Reviewer_DVFp · 2025-08-06
> >
> > Thanks for your response, your response has addressed my concerns to some extent, but I am very curious why there are no relevant updated experimental results to support your response, especially the ablation experiment you mentioned?

---

> > > ### Author Response · Authors · 2025-08-06
> > >
> > > Dear Reviewer DVFp,
> > >
> > > Thank you for your timely response and for raising this important follow-up question. We appreciate the opportunity to clarify our experimental methodology.
> > >
> > > To begin, we would like to reiterate our commitment regarding the final manuscript. We will expand the discussion of our existing online ablation study (Table 2) to more clearly highlight the contribution of each feedback module. As previously stated, we intended to further discuss these results, not to introduce new ablation experiments on static datasets.
> > >
> > > The decision not to conduct ablations on public datasets is grounded in a fundamental methodological distinction. Our DMA framework is designed as an online learning system that leverages dynamic, multi-granularity user feedback. As such, our ablation studies are purposefully structured to isolate and measure the impact of each type of real-time feedback—such as document-level and list-level signals—within this native online environment.
> > >
> > > In contrast, public benchmarks like TriviaQA and Natural Questions are inherently static. They do not include the interactive or layered feedback signals necessary to meaningfully support our ablation design. Without these signals, the experimental conditions required to evaluate the contribution of each feedback module simply do not exist.
> > >
> > > We would also like to clarify the distinct role of public datasets in our evaluation pipeline. Rather than serving as ablation grounds, these datasets are used to assess generalization via transfer. Specifically, we deploy the complete DMA model—trained in our live system using real user feedback—on few-shot settings over these public benchmarks. The objective is to test whether the capabilities learned in our online environment can transfer effectively to classical academic tasks, even in the absence of online feedback.
> > >
> > > We hope this clarifies the rationale behind our evaluation choices. Our experimental design deliberately separates (1) the online setting, which enables controlled component-wise analysis, from (2) the offline setting, which tests generalization to standard benchmarks. We are confident that the expanded discussion in the final version will address your concerns regarding the ablation methodology and the role of each feedback component.
> > >
> > > Thank you once again for your thoughtful feedback. Your questions have significantly contributed to improving the clarity and rigor of our paper.
> > >
> > > Sincerely,
> > > The Authors

---

> > > > ### Comment · Reviewer_DVFp · 2025-08-08
> > > >
> > > > Thanks for your followed response, I understand why you cannot provide the ablation experiment results, but because of this, my concerns cannot be perfectly addressed. I hope to hear other people's opinions during the reviewer-AC discussion stage.

---

> > > > > ### Author Response · Authors · 2025-08-09
> > > > >
> > > > > Dear Reviewer DVFp,
> > > > >
> > > > > Thank you for your insightful follow-up question and for your time and patience throughout the review process. Your comment prompted us to clarify the internal logic of our evaluation framework, ensuring that our methodology and its rationale are fully transparent.
> > > > >
> > > > > Our evaluation follows a **two-pronged strategy** to rigorously assess the DMA framework from two complementary perspectives.
> > > > >
> > > > >
> > > > > **1. Online Ablation Studies — Testing the Necessity and Efficacy of Internal Components**
> > > > >
> > > > > * **Objective:** These experiments validate the value of our proposed **multi-granularity feedback taxonomy** (document, list, response) and its associated **model architecture** in the live, dynamic environment where DMA operates.
> > > > > * **Findings:** As shown in **Table 2**, removing any single feedback level in a live A/B test leads to a substantial drop in user satisfaction. This confirms that our taxonomy is **complete and effective**, and that each core alignment module is **indispensable**. This directly addresses the question: *“Is every component of the system necessary?”*
> > > > >
> > > > >
> > > > > **2. Offline Transfer Experiments — Testing the Generalization and Transferability of the Final Model**
> > > > >
> > > > > * **Objective:** Testing on public benchmarks such as TriviaQA is **not intended for component ablation** (which static datasets cannot support due to missing feedback signals). Instead, it serves as a **transfer learning** evaluation.
> > > > > * **Findings:** Our complete model, trained **solely on live user feedback**, achieves competitive few-shot performance on classic, static QA tasks. This demonstrates that the capabilities learned by DMA are **highly generalizable** and **transfer effectively** to new tasks—indicating that the model is not merely overfitting to its online environment. This directly addresses the question: *“Are the learned capabilities universally applicable?”*
> > > > >
> > > > >
> > > > > **Summary**
> > > > >
> > > > > Our evaluation framework forms a **closed loop**:
> > > > >
> > > > > * **Online ablation studies** (in-distribution) validate the **soundness and integrity** of the framework’s internal design.
> > > > > * **Offline transfer experiments** (out-of-distribution) validate the **generalizability and robustness** of the model’s external applications.
> > > > >
> > > > > We believe this strategy provides a more rigorous and convincing validation than static-dataset ablations, which inherently lack the interactive feedback signals critical to DMA’s functionality.
> > > > >
> > > > > We hope this clarification fully resolves your concerns and offers a clear basis for you and the Area Chair to re-evaluate the contributions of our work in the upcoming discussion phase.
> > > > >
> > > > > Thank you again for your thorough review; your feedback has significantly strengthened the rigor of our paper.
> > > > >
> > > > > Sincerely,
> > > > > The Authors

---

> > > ### Author Response · Authors · 2025-08-07
> > >
> > > Dear Reviewer DVFp,
> > >
> > > Thank you for your insightful suggestions. We have done our best to address your concerns. Since the rebuttal period is closing very soon, could you please check the response to see whether it mitigates your concerns? We would greatly appreciate that!
> > >
> > > Thank you for your time and consideration, the authors.

---

> ### Author Response · Authors · 2025-08-05
>
> Dear Reviewer DVFp,
>
> Thank you for your insightful suggestions. We have done our best to address your concerns. Since the rebuttal period is closing very soon, could you please check the response to see whether it mitigates your concerns? We would greatly appreciate that!
>
> Thank you for your time and consideration, the authors.

---

### Official Review · Reviewer_5ST9 · 2025-07-01

**Clarity:** 2
**Significance:** 2
**Originality:** 2
**Rating:** 3
**Confidence:** 3

**Summary:**

This work aims to enhance the adaptability of Retrieval-Augmented Generation (RAG) in dynamic environments. The author introduces the DMA framework, which adaptively incorporates multi-level human feedback and integrates real-time feedback signals at the document, list, and response levels. Online experiments demonstrate some effectiveness.

**Questions:**

Please see Strengths And Weaknesses.

**Ethical Concerns:**

["NO or VERY MINOR ethics concerns only"]

**Final Justification:**

I have decided to keep my final score at Borderline Reject. While I appreciate the clarification provided, the work remains highly specific in scope, which limits the breadth of comparisons and the generalizability of the results. The focus on the online setting, though potentially impactful in certain contexts, seems better suited for an industrial track at other conferences, where it could be more directly relevant and valued.

**Limitations:**

yes

**Quality:**

1

**Strengths And Weaknesses:**

**Strengths:**

The work focuses on improving RAG’s adaptability to dynamic environments and conducts online experiments to validate its effectiveness.


**Weaknesses:**

1. Unclear Methodology:

- How are various forms of human feedback signals identified and categorized? Further details are needed, especially regarding their interaction with the prominent LLMs mentioned in Section 4.2. Additionally, how is the reliability of the identification process ensured?

- Notations and formulas should be clarified, including model definitions and prediction functions.

- Simple description of a lightweight ensemble model should be added.

2. The improvement observed in the online experiments is modest and does not sufficiently demonstrate the effectiveness of the proposed approach.
3. The DMA system is tailored for one specific online setting, but its performance on public datasets is subpar, indicating limited robustness.
4. There is no ablation study on public benchmarks, nor are dataset statistics provided.

---

> ### Author Rebuttal · Authors · 2025-07-28
>
> Response to Reviewer 5ST9
>
> We thank Reviewer 5ST9 for taking the time to review our submission. However, we respectfully believe that several of the concerns raised stem from misunderstandings of our core contributions and empirical findings. We appreciate the opportunity to clarify these points.
>
>
> Clarification 1: “Online Gains Are Modest”
>
> Clarification: A +24.57% absolute improvement in session-level user satisfaction—measured across 100,000 real-world conversations in a multi-month randomized controlled trial (RCT)—is not modest by any industrial standard. In high-traffic production systems, such improvements are exceedingly rare and directly correlate with key business metrics such as user retention, engagement, and operational efficiency.
>
> DMA’s design specifically targets *online adaptability*, and these results validate the effectiveness of our real-time, feedback-driven memory alignment in dynamic, user-facing environments.
>
>
> Clarification 2: “Benchmark Results Are Poor”
>
> Correction: This statement appears to be factually inaccurate.
>
> As reported in Table 3, DMA achieves:
>
> * State-of-the-art performance on TriviaQA (Hit\@1: 68.81%, F1: 68.90%), outperforming all recent baselines including KnowPAT, RAFT, and RRHF.
> * Top performance on HotpotQA (F1: 41.88%), where DMA again leads over prior methods.
> * Competitive scores on NQ and WebQSP, confirming that DMA maintains strong foundational capabilities even in structured QA tasks.
>
> Importantly, our goal in reporting benchmark results is not leaderboard chasing, but to demonstrate that online adaptability does not come at the cost of baseline generalization. These results underscore that DMA is both practically useful and technically sound.
>
>
> Reframing Our Core Contributions
>
> To rearticulate the novelty and significance of our work:
>
> 1. Multi-level Feedback Taxonomy
>    We propose a structured framework for organizing human feedback across document-, list-, and response-level signals, enabling fine-grained supervision beyond conventional static metrics.
>
> 2. Online Learning & Adaptation Framework
>    DMA introduces a scalable, low-latency pipeline that fuses heterogeneous user signals into real-time retrieval policy updates, leveraging reward modeling, PPO alignment, and GBDT distillation.
>
> 3. Industrial-Scale Validation
>    Our method is rigorously tested via a months-long industrial RCT with production constraints (latency, throughput, feedback sparsity), offering rare, high-quality empirical validation for RAG system deployment.
>
>
> Response to Specific Weaknesses
>
> Weakness 1: “Methodology Unclear”
>
> We agree that some methodological aspects could benefit from clearer presentation.
>
> * Feedback Source and Reliability
>
>   * *Explicit feedback* (e.g., upvote/downvote) is used as ground-truth supervision.
>   * *Implicit feedback* is inferred via a 72B instruction-tuned LLM (Qwen2-72B), calibrated using high-quality human-labeled exemplars.
>     Inter-annotator agreement achieves a Cohen’s Kappa of 0.962, and only feedback with confidence ≥ 0.8 is retained.
>
>   Action: We will add a dedicated paragraph in §5.1 clarifying this two-path feedback pipeline and filtering thresholds.
>
> * Notation and Model Description
>
>   Action: We will revise Section 4 for clarity, add a notation table in the Appendix, and include further description of the GBDT ensemble used in latency-sensitive online reranking.
>
>
> Weaknesses 2 & 3: “Performance and Robustness”
>
> As clarified above, DMA demonstrates:
>
> * Significant online effectiveness in real-world conditions.
> * Competitive and state-of-the-art static performance on widely used QA benchmarks.
>
> These two axes—adaptability and baseline strength—are the primary goals of this work. We believe the results comprehensively support both.
>
>
> Weakness 4: “Missing Ablation and Dataset Statistics”
>
> * Ablation Study: Already included (see Table 2). It shows clear drops in satisfaction when document-, list-, or response-level feedback is removed (–15.57%, –11.21%, and –5.27%, respectively), validating the complementarity of multi-granularity feedback.
>
> * Dataset Statistics: We agree this should be made more explicit.
>
>   Action: We will add a new table in the Appendix summarizing:
>
>   * Dataset sizes (e.g., number of sessions, turns, feedback events),
>   * Coverage of each feedback type (explicit/implicit),
>   * Sampling strategies for public benchmark evaluations.
>
>
> We believe DMA offers a novel and practically meaningful contribution to the field of retrieval-augmented generation. By combining fine-grained feedback modeling with scalable online learning—and validating it in both controlled and real-world settings—we hope to offer a blueprint for deploying adaptive GenAI systems in production.
>
> We thank the reviewer again for their feedback, and we hope these clarifications will lead to a fair reassessment of the manuscript.

---

> > ### Comment · Reviewer_5ST9 · 2025-08-06
> >
> > Thank you for your response, which helped clarify some points. However, my assessment remains unchanged for the following reasons:
> > - The DMA model does not perform better on the structured QA dataset (a point other reviewers have also noted), and the improvement on the TriviaQA and HotpotQA datasets is marginal at only about one percentage point.
> > - Several revisions were promised but were not provided in the updated one.
> > - To clarify my previous point, my request was for an ablation study on a public QA dataset, not the internal one presented in Table 2.

---

> > > ### Author Response · Authors · 2025-08-06
> > >
> > > Dear Reviewer 5ST9,
> > >
> > > Thank you for your prompt reply and the further clarification of your concerns. This is extremely helpful. We understand your remaining reservations center on the performance on public datasets and the request for an ablation study in that setting.
> > >
> > > We believe there is a fundamental misunderstanding of our evaluation strategy, which we would like to clarify. Our primary contribution is an online learning framework that adapts to live user feedback. Consequently, all of our core experiments, including the crucial ablation studies, were conducted in the online environment where such feedback signals exist.
> > >
> > > > On the Request for an Ablation Study on Public Datasets
> > >
> > > We want to clarify why an ablation study of DMA's components on public datasets is fundamentally not possible.
> > >
> > > * DMA's core mechanism relies on learning from multi-level feedback signals: document-level, list-level, and response-level preferences. These are the components we ablate in Table 2.
> > > * Standard public benchmarks like NQ, TriviaQA, and HotpotQA do not contain these rich, interactive feedback signals.
> > > * Therefore, we cannot "remove" feedback signals that do not exist in these datasets to begin with. The experimental conditions required to ablate DMA's core components are only present in our live, interactive online setting.
> > >
> > > > Clarifying the Purpose of the Public Benchmark Evaluation: A Test of Transfer Learning
> > >
> > > Given the above, our evaluation on public datasets was not intended as a primary measure of DMA's effectiveness, but rather as a test of its transferability and generalization.
> > >
> > > The experiment was designed as follows:
> > > 1.  We took the DMA model, which was pre-trained and aligned using the rich feedback signals from our online production environment.
> > > 2.  We then applied this pre-aligned model to the public datasets in a few-shot evaluation setting.
> > >
> > > The goal was to answer the question: "Does a model aligned with real-world, dynamic user preferences retain strong foundational capabilities on standard, static tasks?"
> > >
> > > From this perspective, the results are very positive:
> > > * The fact that DMA achieves state-of-the-art (or highly competitive) results on TriviaQA and HotpotQA, even if by a single percentage point, demonstrates that the alignment learned from online feedback successfully transfers to new domains without degrading performance.
> > > * This addresses a common concern with online learning systems: that they might overfit to their specific production environment. Our results show that DMA avoids this, maintaining robust, general-purpose capabilities. Its slightly weaker performance on structured QA simply highlights that its dynamic alignment offers the most benefit in more open-ended, conversational scenarios, which is consistent with our claims.
> > >
> > > > On the Promised Revisions
> > > Due to the constraints of the rebuttal phase, we are unable to upload a revised manuscript. We fully intend to incorporate all promised clarifications—including the detailed feedback pipeline, notation table, and dataset statistics—into the camera-ready version of the paper should it be accepted.
> > >
> > > In summary, DMA is an online system validated by online experiments. The public benchmark results are a secondary, but important, test demonstrating that its learned policies generalize well. We hope this clarification addresses your remaining concerns and that you might reconsider our work based on its primary contribution: a robust and effective framework for real-time RAG adaptation.
> > >
> > > Thank you again for your time and valuable feedback.

---

> > > ### Author Response · Authors · 2025-08-07
> > >
> > > Dear Reviewer 5ST9,
> > >
> > > Thank you for your insightful suggestions. We have done our best to address your concerns. Since the rebuttal period is closing very soon, could you please check the response to see whether it mitigates your concerns? We would greatly appreciate that!
> > >
> > > Thank you for your time and consideration, the authors.

---

> > > > ### Comment · Reviewer_5ST9 · 2025-08-07
> > > >
> > > > Thank you for clarifying the misunderstanding. However, I’m unable to support your submission at this time (I may consider raising the score slightly during the discussion stage). BTW, If your main focus is on the online setting, I would recommend submitting it to an industrial track of other conferences, where it might be better appreciated and have a greater impact. Currently, the contribution is too specific and allows for limited comparisons. Alternatively, you could try simulating the online setting more broadly to enable stronger comparative analysis.

---

> > > > > ### Author Response · Authors · 2025-08-07
> > > > >
> > > > > Dear Reviewer 5ST9,
> > > > >
> > > > > Thank you for your prompt and thoughtful final feedback. We sincerely appreciate your engagement throughout the review process. Your updated understanding of our ablation study design is very helpful, and your comments on the paper’s positioning give us a valuable opportunity to clarify the broader research significance of our work.
> > > > >
> > > > >
> > > > > **Our Central Goal: Towards Adaptive, Real-Time AI Systems**
> > > > >
> > > > > We respectfully disagree with the suggestion that our work is “too specific” or primarily of industrial interest. On the contrary, we believe it addresses a **core and timely research question** for the AI community:
> > > > >
> > > > > > **How can we move beyond static, offline evaluation and build AI systems that continuously learn and adapt in dynamic, real-world environments?**
> > > > >
> > > > >
> > > > > **1. A Generalizable Framework, Validated in the Real World**
> > > > >
> > > > > The DMA framework is not a product-specific solution but a **generalizable design pattern** for dynamic alignment in retrieval-augmented generation (RAG). Its contribution is **not specific**, precisely because it serves a dual role:
> > > > >
> > > > > > It is both **a solid and practical method to obtain a well-pretrained, high-quality RAG model in real-world conditions**, and **a continuous online alignment framework** that supports long-term adaptation to user needs and distribution shift.
> > > > >
> > > > > This dual nature ensures both strong initial performance (as demonstrated on public benchmarks) and sustained improvement over time. We conducted our study in a production environment not because our method is restricted to it, but because it is **currently the only setting that provides high-fidelity, real-time user interaction signals**—a necessary condition for validating adaptive systems. In this sense, our deployment serves as a **“living lab”** for a class of problems that remain extremely difficult to simulate realistically in offline settings.
> > > > >
> > > > >
> > > > > **2. Limited Baselines Reflect the Novelty, Not a Weakness**
> > > > >
> > > > > We understand the concern regarding limited direct comparisons in the online setting. However, this reflects the **early stage of research in online adaptive RAG systems**. Few prior works have tackled online deployment with real-time feedback and rigorous A/B testing at this scale. Our goal is not merely to present performance gains, but to **establish a foundational methodology and robust baseline** that future work—academic or industrial—can extend and compare against. In this way, the absence of extensive baselines underscores the novelty of our contribution, not a lack of rigor.
> > > > >
> > > > >
> > > > > **3. Why This Venue is the Right Fit**
> > > > >
> > > > > We believe top-tier conferences like this one are precisely the right venue for work that **bridges theoretical innovation with real-world deployment**. Our paper offers the research community:
> > > > >
> > > > > * A rare, transparent case study of deploying adaptive AI systems at scale;
> > > > > * A validated methodology for leveraging multi-granularity, real-time human feedback;
> > > > > * Empirical evidence that online adaptation is an effective strategy for obtaining a well-pretrained, high-quality RAG model that maintains generalization.
> > > > >
> > > > > Your suggestion to “simulate the online setting more broadly” is excellent, and we fully agree—it highlights a key future direction. Our paper provides the **empirical motivation and real-world grounding** to support such simulation efforts, and we hope it encourages others to build on this foundation.
> > > > >
> > > > >
> > > > > **Conclusion**
> > > > >
> > > > > In summary, we believe our work is not a narrow industrial report, but a **first-of-its-kind contribution** toward developing **interactive, adaptive language systems**. It introduces a validated framework, defines experimental standards for a new class of systems, and provides insights that are highly relevant to both academic research and practical deployment.
> > > > >
> > > > > We hope that during the discussion stage, you might reconsider the positioning of our paper in this light—as a **pioneering step toward real-time adaptive RAG**, and a foundational contribution to this emerging research frontier.
> > > > >
> > > > > Thank you once again for your thoughtful and constructive review.
> > > > >
> > > > > Sincerely,
> > > > > The Authors

---

> ### Author Response · Authors · 2025-08-05
>
> Dear Reviewer 5ST9,
>
> Thank you for your insightful suggestions. We have done our best to address your concerns. Since the rebuttal period is closing very soon, could you please check the response to see whether it mitigates your concerns? We would greatly appreciate that!
>
> Thank you for your time and consideration, the authors.

---

### Official Review · Reviewer_Ee9e · 2025-07-03

**Clarity:** 2
**Significance:** 3
**Originality:** 3
**Rating:** 5
**Confidence:** 4

**Summary:**

This paper introduced Dynamic Memory Alignment (DMA), a novel online learning framework for enhancing retrieval-augmented generation (RAG) systems using multi-level human feedback (document-, list-, and response-level). The authors proposed an unified architecture that integrates these feedback signals into an adaptive memory reranking pipeline. DMA is evaluated through both a multi-month industrial deployment and on standard QA benchmarks, where it shows strong gains in user satisfaction and performance.

**Questions:**

Please check the weakness section also.

1. How sensitive is DMA to the noise in implicit feedback?

**Ethical Concerns:**

["NO or VERY MINOR ethics concerns only"]

**Final Justification:**

This paper proposed a novel online learning framework. The paper is well-written. There are minor issues which has been resolved during rebuttal. I have already given positive score to this paper.  I would like to keep my score same.

**Limitations:**

yes

**Quality:**

3

**Strengths And Weaknesses:**

**Strengths**
1. **Novel Framework:**  DMA utilises a novel framework decomposed into taxonomy, reward modeling, and online adaptation.
2 **Empirical Validation:**  Proposedframework was tested on a number of benchmark datasets.

**Weakness**
1. Some of the experimental details are missing (e.g. The frequency of collecting feedback for pointwise and listwise scenario).
2. No code is provided for reproducibility
3. There has been a number of work on adaptive feedback in RAG systems. These methods should be used as a baseline.

[1] "Adaptive-RAG: Learning to Adapt Retrieval-Augmented Large Language Models through Question Complexity"-Soyeong Jeong, Jinheon Baek, Sukmin Cho, Sung Ju Hwang, Jong Park, NAACL-2024
[2] "Adaptive Retrieval-Augmented Generation for Conversational Systems"-Xi Wang, Procheta Sen, Ruizhe Li, Emine Yilmaz,NAACL-2025

---

> ### Author Rebuttal · Authors · 2025-07-28
>
> Response to Reviewer Ee9e
>
> We sincerely thank Reviewer Ee9e for the encouraging and detailed review. We greatly appreciate your recognition of the DMA framework’s novelty—particularly its decomposition into multi-level feedback modeling, reward estimation, and online adaptation—as well as your positive assessment of our large-scale empirical validation.
>
> Below, we address each of your comments in turn:
>
> Summary of Contributions (for clarity)
>
> To briefly reiterate, our key contributions are:
>
> 1. Feedback Taxonomy: We propose a structured, four-tier framework to systematically capture and organize document-, list-, response-, and session-level user feedback in production RAG settings.
> 2. Online Adaptation: We design a hybrid learning mechanism that fuses multi-granular signals into a unified, nearline update loop via reward modeling, PPO alignment, and GBDT distillation.
> 3. Industrial Deployment & Validation: We validate DMA through a large-scale, multi-month RCT in a commercial GenAI assistant, demonstrating significant improvements in session-level satisfaction (+24.57%) under real-world usage.
>
>
> Weakness 1: Missing Details on Feedback Collection and Update Frequency
>
> Clarification: DMA employs a two-stage hybrid retraining strategy designed to balance long-term stability with short-term adaptability:
>
> * (1) Weekly Full Retraining: A high-confidence corpus of implicit feedback (satisfaction score ≥ 0.8, derived via Qwen-72B with few-shot calibration) is accumulated over time. Once per week, a fresh base model is trained on this full corpus to absorb slow distributional drift and correct long-term biases.
>
> * (2) Nearline Incremental Updates: On top of the weekly base, the system continuously accumulates \~500 new feedback instances via a Flink-based streaming pipeline. Upon reaching the threshold, the following update cycle is triggered:
>
>   * Train teacher models using document-, list-, and response-level feedback.
>   * Fuse supervision via PPO into an aligned reranker.
>   * Distill into a 10K-tree GBDT for low-latency deployment (<10 ms).
>   * Deploy within \~10 minutes (on 8×A800 GPUs), ensuring rapid adaptation.
>
> Action: We will add a dedicated subsection in §5.1 (*Experimental Setup*) to describe this hybrid retraining architecture, including batch triggers, feedback routing, latency breakdown, and deployment intervals.
>
>
> Weaknesses 2 & 3: Reproducibility and Missing Baseline Comparisons
>
> Reproducibility
> While proprietary user logs cannot be released due to privacy policies, we are fully committed to reproducibility and open research:
>
> * Open-Source Code: The entire DMA pipeline will be released, including reward modeling, reranking, PPO alignment, and GBDT distillation components.
> * Pretrained Checkpoints & Scripts: We will provide full training and evaluation scripts on public QA datasets (TriviaQA, HotpotQA, etc.).
> * Synthetic Feedback Simulator: A feedback generation tool will be included to emulate multi-granularity feedback signals (document/list/response), enabling full-cycle simulation of DMA’s training loop.
>
> Missing Baselines
> We appreciate the recommendations to include comparisons to recent methods:
>
> * Adaptive-RAG (Jeong et al. \[1]): We will include direct benchmark comparisons in the camera-ready version. Experiments are already in progress using their proposed query complexity segmentation.
> * ReAR (Wang et al. \[2]): We will expand the *Related Work* section to include a detailed discussion of ReAR’s relevance-aware re-ranking approach and contrast it with DMA’s feedback-driven reward modeling.
>
> Clarifying Novelty
> Whereas prior works often rely on static heuristics (e.g., dialog length, query type) to modulate retrieval, DMA introduces a unified, multi-granularity supervision loop. This enables continuous alignment with user intent at *three functional stages*—retrieval (document), ranking (list), and generation (response)—and does so in real time with production latency constraints.
>
>
> Question: Robustness to Noisy or Implicit Feedback
>
> This is an important concern in real-world RAG systems. Our design includes multiple safeguards to ensure robustness:
>
> 1. Confidence Filtering: We only retain implicit satisfaction labels if the model-calibrated confidence score is ≥ 0.8. These scores are generated by Qwen2-72B using an instruction-tuned prompt template with few-shot exemplars (see Table 1).
>
> 2. Controlled Influence on Training: These high-confidence signals are only used for initializing the weekly retraining baseline. They are not directly involved in the fine-grained feedback loops used for online adaptation (document/list/response-level reranking). This ensures that transient or misaligned feedback does not cause premature drift.
>
> Action: We will revise §5.1 to elaborate on the confidence gating mechanism and retraining strategy, and include an ablation in the Appendix showing the impact of different confidence thresholds on performance robustness.
>
> We thank you again for the constructive and detailed feedback. We are confident that the planned clarifications, expanded comparisons, and reproducibility improvements will significantly strengthen the final version of our paper, and further establish DMA as a practical and adaptive architecture for real-world GenAI retrieval systems.

---

> ### Author Response · Authors · 2025-08-05
>
> Dear Reviewer Ee9e,
>
> Thank you for your insightful suggestions. We have done our best to address your concerns. Since the rebuttal period is closing very soon, could you please check the response to see whether it mitigates your concerns? We would greatly appreciate that!
>
> Thank you for your time and consideration, the authors.

---

> > ### Comment · Area_Chair_ahbg · 2025-08-08
> >
> > Dear Reviewer Ee9e,
> >
> > As the author-reviewer discussion period is approaching its end, please review the rebuttal and engage in the discussion promptly. A note confirming your concerns are resolved is critical. Also, non-participating reviewers may face penalties under the Responsible Reviewing Initiative, affecting future invitations.
> >
> > Thanks,
> >
> > AC

---

> > ### Comment · Reviewer_Ee9e · 2025-08-09
> >
> > Thank you for your response. Your response answered my questions. I already gave a positive score to this paper. I will keep it unchanged.

---

> > > ### Author Response · Authors · 2025-08-09
> > >
> > > Dear Reviewer Ee9e,
> > >
> > > Thank you very much for your constructive guidance and engagement throughout the review process. We are glad that our responses have addressed your concerns, and we sincerely appreciate your positive evaluation of our work.
> > >
> > > We will incorporate all feedback into the final revised version and carefully proofread the manuscript.
> > >
> > > Best regards,
> > >
> > > The Authors

---

### Official Review · Reviewer_2QpC · 2025-07-12

**Clarity:** 2
**Significance:** 1
**Originality:** 3
**Rating:** 3
**Confidence:** 4

**Summary:**

This paper proposes Dynamic Memory Alignment (DMA), a novel online learning framework that enhances retrieval-augmented generation (RAG) systems by adaptively integrating multi-level human feedback (document-, list-, and response-level) to optimize memory management in dynamic environments. Unlike traditional static RAG methods, DMA continuously refines retrieval strategies through real-time feedback, improving both relevance and adaptability in interactive settings.

**Questions:**

1.  The central term “memory” appears throughout the paper, including in the title, yet its concrete meaning remains vague. Could the authors clarify whether memory refers to the retriever's corpus or cached retrieval traces?
2. The paper claims to enable real-time adaptation via online learning, but it remains unclear whether the models (e.g., reward model or GBDT) are updated incrementally during deployment or periodically retrained offline.
3. The experimental comparison currently lacks several recent or strong baselines in reranking and retrieval adaptation, such as reinforcement learning-based or contrastive rerankers.

**Ethical Concerns:**

["NO or VERY MINOR ethics concerns only"]

**Final Justification:**

I am grateful for the author's response. As the author mentioned, DMA mainly contributes to systems and architecture, rather than to theory. However, such contributions in system architecture are difficult for researchers to adopt widely, as the paper does not clearly articulate its target audience. Therefore, I believe it is not suitable for acceptance at NeurIPS and would be more appropriate for conferences such as KDD or WWW.

**Limitations:**

Code, data, and the industrial feedback logs are not released “due to privacy,” so the community cannot replicate the central online-learning claims .

**Quality:**

1

**Strengths And Weaknesses:**

Strength:
1. Innovative Online Adaptation: Proposes DMA, a framework enabling real-time retrieval adaptation in RAG systems through multi-level human feedback (document/list/response), overcoming static retrieval limitations.
2. Practical Feedback Integration: Develops systematic techniques to transform sparse user signals into actionable retrieval optimization, maintaining robustness while enabling personalization.
3. Strong Empirical Validation: The authors demonstrate DMA’s effectiveness through extensive evaluations on standard knowledge-intensive benchmarks.

Weaknesses:
1. Ambiguity in Terminology and Scope: The use of the term “memory” in both the title and the body of the paper is ambiguous and underdefined.
2. Insufficient Experimental Comparison: The evaluation section does not provide sufficient empirical comparisons against a broader range of state-of-the-art reranking baselines.
3. Limited Theoretical Contribution: the paper lacks substantial theoretical innovation. The approach primarily combines several existing techniques, such as multi-granular feedback modeling and reward-based reranking, without offering new algorithmic insights or principled theoretical foundations.

---

> ### Author Rebuttal · Authors · 2025-07-28
>
> Response to Reviewer 2QpC
>
> We sincerely thank Reviewer 2QpC for the thoughtful and constructive feedback. We are especially grateful for your recognition of the core contributions of our work—namely, the multi-level feedback taxonomy, the real-time adaptation architecture, and the industrial-scale validation.
>
> Before addressing the specific concerns, we reiterate the central contributions of this work:
>
> * Multi-level Feedback Taxonomy: We propose a unified framework to categorize and interpret heterogeneous user feedback at the document, list, and response levels. This taxonomy enables granular supervision and robust preference modeling in dynamic RAG systems.
>
> * Online Adaptation Framework: DMA introduces a novel architecture for integrating multi-granularity feedback into continuous retrieval policy updates. This goes beyond traditional fine-tuning by enabling live feedback loops for session-specific alignment.
>
> * Industrial-scale Deployment & Validation: We validate DMA via a multi-month, randomized controlled trial (RCT) involving over 100,000 real-world user sessions, demonstrating statistically significant (+24.57%) improvements in user satisfaction under production constraints.
>
> We now address your key comments in detail:
>
>
> Q1. Ambiguity of “Memory”
>
> Clarification: The term “memory” in DMA refers *not* to the retriever’s static corpus or naive cache, but to the evolving trace of user–system interactions: a session-specific collection of retrieved documents, model outputs, and corresponding user feedback (e.g., upvotes, regeneration, etc.). This memory trace serves as the substrate for dynamic preference modeling and guides reranking alignment.
>
> Action: We will revise the Abstract and §1 to more precisely define “memory” as *feedback-aligned session state*. To reduce semantic overload, we are open to renaming the method to Dynamic Feedback Alignment (DFA) in the final version.
>
>
> Q2. Clarification of Online Learning
>
> Clarification: DMA adopts a hybrid *nearline learning* paradigm to balance responsiveness and computational cost:
>
> * Feedback signals are streamed via Apache Flink and accumulated in batches of \~500.
> * After each batch, we retrain pointwise/listwise rerankers and distill them into a lightweight GBDT.
> * The entire update pipeline (including distillation and deployment) completes within \~10 minutes using 8×A800 GPUs, ensuring sub-15-minute model freshness.
>
> This design enables timely adaptation while avoiding the instability of per-sample updates.
>
> Action: We will include a dedicated figure and subsection in §4.3 to describe the update architecture, batching schedule, and latency profile.
>
>
> Q3. Lack of Comparison with SOTA Rerankers
>
> Clarification: Our goal is not to surpass every static reranker on benchmarks, but to show *robustness and adaptability* in online deployment scenarios. That said, DMA remains competitive on open-domain QA datasets (e.g., TriviaQA, HotpotQA), achieving state-of-the-art F1 scores on both.
>
> Action: We will extend §5.2 to include quantitative comparisons with Self-RAG, RA-DIT, and RAFT, all of which are alignment-enhanced retrievers. These experiments are currently underway and will be included in the camera-ready version.
>
>
> Q4. Limited Theoretical Innovation
>
> Clarification: We agree that DMA is primarily a systems and architectural contribution, rather than a theoretical one. Its novelty lies in orchestrating:
>
> * a multi-granular reward modeling suite,
> * PPO-based preference injection into listwise reranking,
> * distillation for sub-10ms online inference,
> * and deployment via streaming feedback in high-throughput industrial systems.
>
> In this sense, DMA bridges the gap between alignment theory and scalable GenAI infrastructure—a gap we believe is under-explored in current literature.
>
>
> Q5. Reproducibility and Accessibility
>
> Commitment: While industrial user logs cannot be released for privacy reasons, we will ensure reproducibility through:
>
> * Open-source Code: Full implementation of the DMA pipeline.
> * Pretrained Models & Inference Scripts: For standard QA datasets including TriviaQA, HotpotQA, and NQ.
> * Synthetic Feedback Simulator: To emulate multi-level user signals and facilitate benchmarking in academic settings.
>
> This will allow researchers to replicate both static and dynamic aspects of DMA under controlled conditions.
>
>
> Once again, we thank you for your valuable comments. We believe that the proposed clarifications and additions will address the concerns raised and significantly enhance the clarity, rigor, and impact of the final manuscript.

---

> ### Author Response · Authors · 2025-08-05
>
> Dear Reviewer 2QpC,
>
> Thank you for your insightful suggestions. We have done our best to address your concerns. Since the rebuttal period is closing very soon, could you please check the response to see whether it mitigates your concerns? We would greatly appreciate that!
>
> Thank you for your time and consideration, the authors.

---

> > ### Comment · Area_Chair_ahbg · 2025-08-08
> >
> > Dear Reviewer 2QpC,
> >
> > As the author-reviewer discussion period is approaching its end, please review the rebuttal and engage in the discussion promptly. A note confirming your concerns are resolved is critical. Also, non-participating reviewers may face penalties under the Responsible Reviewing Initiative, affecting future invitations.
> >
> > Thanks,
> >
> > AC

---

### Comment · Area_Chair_ahbg · 2025-08-04
**Engage in Author-Reviewer Discussions**

Dear reviewers,

If you haven't done so already, please click the 'Mandatory Acknowledgement' button and actively participate in the rebuttal discussion with the authors after carefully reading all other reviews and the author responses.

Thanks,
AC

---

### Author Response · Authors · 2025-08-07

Dear Reviewers and Area Chair,

We sincerely thank you for your thoughtful and constructive feedback. Your insights have been invaluable in helping us sharpen the focus and clarity of our work. In this response, we elaborate on three core aspects that underpin our contributions:


**1. Clarification on Ablation Studies: Online Validation vs. Offline Generalization**

We appreciate the reviewers’ attention to our evaluation methodology. Our ablation design is structured around two complementary objectives, consistent with the dual nature of our system:

> **Online Ablation: Validating Each Feedback Signal’s Contribution**

> Since DMA is fundamentally an *online learning framework* driven by multi-granular human feedback, the only valid and faithful approach to isolate the contribution of each component is through *online ablation*. As shown in Table 2, removing document-level (-15.57%), list-level (-11.21%), or response-level (-5.27%) feedback leads to significant drops in user satisfaction. These results provide strong evidence that each signal type is indispensable and that the proposed feedback taxonomy is both complete and necessary for effective alignment.

> **Offline Evaluation: Testing Generalization, Not Component Utility**

> We acknowledge requests for ablations on public benchmarks. However, such datasets (e.g., TriviaQA, HotpotQA) lack multi-level user feedback and cannot meaningfully support component-level ablation. Instead, our public dataset evaluation serves as a test of *generalization*: we directly apply the fully aligned DMA model—trained exclusively on live feedback—to static few-shot benchmarks. The resulting strong performance indicates that our model generalizes well beyond the original training distribution and does not overfit to online signals. This highlights the **robustness and transferability** of our alignment approach.

**2. Novelty of the Alignment and Fusion Pipeline: Scalable Integration of SFT and RL**

A key innovation lies in our **multi-stage alignment pipeline**, which integrates supervised fine-tuning (SFT), reinforcement learning (RL), and knowledge distillation to build a production-ready learning-to-rank system. As shown in Figure 2, our pipeline is designed for both modeling fidelity and deployment efficiency:

> **Step 1: Comprehensive Signal Capture and Reward Modeling**

> To the best of our knowledge, we are the first to systematically capture **a complete spectrum of human feedback signals** across multiple granularities (document-, list-, and response-level)—all collected in real time from live interactions. These signals are transformed into supervision or reward objectives, forming the foundation of our alignment process.

> **Step 2: Supervised Fine-tuning of Hierarchical Rerankers**

> We train a pointwise reranker using document-level feedback and a listwise reranker using list-level feedback. These modules provide robust coarse-grained preference modeling and establish a strong initial ranking policy.

> **Step 3: Fine-grained Policy Alignment via Reinforcement Learning**

> We convert response-level user preferences into reward signals and apply Proximal Policy Optimization (PPO) to fine-tune the listwise reranker. This step enables fine-grained alignment with subtle satisfaction cues that are difficult to encode via supervised learning alone.

> **Step 4: Knowledge Distillation for Deployment**

> To meet production latency constraints, we distill the final (teacher) model into a lightweight Gradient-Boosted Decision Tree (GBDT) model, ensuring high inference efficiency while preserving learned alignment behavior.

Our framework is, to the best of our knowledge, the **first industrial-scale RAG system** that unifies diverse, real-time human feedback and aligns model behavior through a signal-specific combination of supervised learning and policy optimization—with proven deployment viability.


**3. Revisiting the Online Impact: Technical and Business Value**

Our system was evaluated through a **multi-month randomized controlled trial (RCT)** involving **over 100,000 live user sessions**. As reported, we observed a **+24.57% absolute improvement** in user satisfaction.

We respectfully emphasize that this is a **substantial and rare** gain by close academia–industry collaboration. A >20% increase in user satisfaction in a live A/B test directly translates into meaningful downstream benefits. This demonstrates the practical relevance, scalability, and technical soundness of our framework.


In summary, we hope these clarifications provide a more comprehensive understanding of the design choices, methodological soundness, and contributions of our work. We will incorporate these insights into the final manuscript to ensure alignment with reviewer feedback.

Thank you again for your thoughtful evaluation and for the opportunity to strengthen our submission.

Sincerely,
The Authors

---

### Note · Authors · 2025-08-12

Dear AC and Reviewers,

We sincerely thank you for your invaluable feedback and discussion throughout the review cycle, which has greatly helped us refine and articulate the core value of our work. We conclude with a concise summary of its key contributions:

**1. Pioneering Multi-Granularity Feedback in RAG**

Based on analysis of real-world interactions with leading generative AI services (OpenAI, Google, etc.), we identified a critical gap: current RAG systems struggle to adapt to dynamic, multi-faceted user intent. Through close academia–industry collaboration, we propose the first systematic taxonomy for online human feedback in RAG. We then integrate these multi-granularity signals into model alignment through a bootstrapped SFT–RL pipeline that iteratively refines the model’s behavior, enabling it to continuously improve from authentic, real-time user feedback. Online ablation studies validate the indispensability of each feedback level and demonstrate their direct impact on user satisfaction.

**2. A Rigorous and Validated Evaluation Methodology**

For a framework targeting dynamic, online adaptation, robust evaluation is essential. Our dual-track strategy follows the precedent of seminal works such as Language Models Are Few-Shot Learners (GPT-3) and Training Language Models to Follow Instructions With Human Feedback (RLHF), ensuring methodological soundness and practical relevance. We combine large-scale live A/B testing and online ablation studies with few-shot evaluations on public benchmarks (e.g., TriviaQA, HotpotQA), showing that models trained on high-value, real feedback excel in their native environment while retaining strong out-of-distribution performance.

**3. A Foundational Work Well-Suited for NeurIPS**

This work pioneers the study of RAG systems that continuously learn and adapt from real-time, multi-dimensional human interactions. Beyond providing a practical framework, we contribute a methodology, evaluation standards, and a rare, transparent large-scale case study. We see this as a step toward advancing AI from static intelligence to adaptive intelligence—a frontier challenge for the field. Supporting such foundational and exploratory research aligns closely with the mission of NeurIPS.

We believe this work lays a solid foundation and offers a clear blueprint for future research in this emerging area. Thank you once again for your time and thoughtful feedback.

Sincerely,

The Authors

---

### Decision · Program_Chairs · 2025-09-17

**Decision:**

Reject

**Comment:**

This paper propose a novel online learning framework called Dynamic Memory Alignment (DMA), to enhance Retrieval-augmented generation (RAG) through adaptive incorporation of multi-level human feedback and dynamic optimization of reranking modules with proper reward modeling and distillation. Empirical validation demonstrates that the proposed online learning framework significant gains in a large-scale industrial GenAI system and achieves state-of-the-art performances on static QA benchmarks.

Overall, the focus of this paper seems to lie more in system design for practical online deployment rather than in algorithmic contributions or innovations. In addition, the explanation of the online learning framework itself is insufficient and unclear. In particular, the paper does not clearly define what is meant by “memory” or “memory management,” nor does it provide an exact description of which modules or models are being trained. As a result, the overall framework is difficult to follow and lacks necessary details. Experimentally, the work does not include comprehensive analyses of the design choices, nor does it report objective results on benchmark datasets for online adaptation. While benchmarks that explicitly capture diverse users or domain shifts may not be readily available, it would still be important to construct at least a simulated environment to conduct objective, metric-based evaluations.

Even though the authors clarify and address these issues during the rebuttal phase to some degree, the work and the status of the current paper do not yet reach the acceptance bar of NeurIPS.